# Effect of New Thiophene-Derived Aminophosphonic Derivatives on Growth of Terrestrial Plants. Part 2. Their Ecotoxicological Impact and Phytotoxicity Test Toward Herbicidal Application in Agriculture

**DOI:** 10.3390/molecules23123173

**Published:** 2018-12-01

**Authors:** Diana Rogacz, Jarosław Lewkowski, Zbigniew Malinowski, Agnieszka Matusiak, Marta Morawska, Piotr Rychter

**Affiliations:** 1Faculty of Mathematics and Natural Science, Jan Długosz University in Częstochowa, 42-200 Częstochowa, 13/15 Armii Krajowej Av., Poland; diana.rogacz@gmail.com; 2Department of Organic Chemistry, Faculty of Chemistry, University of Łódź, Tamka 12, 91-403 Łódź, Poland; zbigniew.malinowski@chemia.uni.lodz.pl (Z.M.); agusk@poczta.fm (A.M.); mz.morawska@gmail.com (M.M.)

**Keywords:** herbicidal activity, persistent weeds, *Aliivibrio fischeri* test, *Heterocypris incongruens* test, ecotoxicology, OECD standard

## Abstract

***Background:*** The aim of this work was to evaluate phytotoxicity of the thiophene derivatives against three persistent weeds of a high degree of resistance (*Galinsoga parviflora* Cav., *Rumex acetosa* L., and *Chenopodium album*) as well as their ecotoxicological impact on *Heterocypris incongruens*. In addition, *Aliivibrio fischeri* was measured. Two of eight described aminophosphonates, namely dimethyl *N*-(2-methoxyphenyl)amino(2-thienyl)methylphosphonate (**2d**) and dimethyl *N*-(*tert*-butyl)- (2-thienyl)methylphosphonate (**2h**), have never been reported before. ***Methods:*** The phytotoxicity of tested aminophosphonates toward their potential application as soil-applied herbicides was evaluated according to the OECD 208 Guideline. Ecotoxicological properties of investigated compounds were made using the OSTRACODTOXKIT^TM^ and Microtox^®^ tests. ***Results:*** Obtained results showed that four aminophosphonates have interesting herbicidal properties and *N*-(2-methylphenyl)amino- (2-thienyl)methylphosphonate (**2a**) was found to kill efficiently the most resistant plant *Chenopodium album*. None of the tested compounds showed important toxicity against *Aliivibrio fischeri*. However, their toxic impact on *Heterocypris incongruens* was significantly elevated. ***Conclusions:*** The aminophosphonate **2a** showed herbicidal potential and it is not toxic against tested bacteria (EC_50_ over 1000 mg/L). It was found to be moderately toxic against ostracods [mortality 48% at 10 mg/kg of soil dry weight (s.d.w.)] and this problem should be solved by the use of the controlled release from a polymeric carrier.

## 1. Introduction

The growing need to increase the food production in agricultural areas stimulates the development of different kinds of pesticides. Among these preparations, herbicides are of interest because of the increasing number of undesirable plant vegetation. Herbicides, which are used currently, are generally more or less safe compounds. However, their widespread and prolonged use and sometimes overdosage resulted in their residues deposition in crops, soil, and land waters, which consequently caused the important human health and environmental hazard [1,2,3,4]. Moreover, such a methodology of application of herbicides results in growing resistance of weeds to a given agent [5,6].

Some of the popular herbicides are reported not to be as safe as they seemed to be and glyphosate is the most distinct example. Numerous reports on its multidimensional toxicity [7,8,9] put a serious question mark over the possibility of its further use.

For this reason, an important gap forms and it creates an opportunity to look for a new, potent, eco-friendly, and selective agent of herbicidal properties.

Sulfur-containing compounds are widely used as plant protection agents and their use and biological properties were recently reported in the field of fungicidal, insecticidal, or herbicidal action [10,11,12]. They are widely used and applied either to weed foliage or to the soil where they are absorbed by roots and/or shoots of emerging seedlings. This problem has been discussed in our previous report, which appeared in the same journal [13]. In this paper [13], we reported the preliminary evaluation of aminophosphonates bearing a 2-substituted thiophene moiety as potential soil-applied herbicides for agricultural/horticultural purposes. We found that these compounds are evidently toxic for dicotyledonous radish and not so harmful for monocotyledonous oat and that their herbicidal efficiency is stronger for the *N*-methylphenyl aminophosphonates substituted with a methyl group at *ortho* and *meta* position in the phenyl ring [13].

The present paper describes investigations, which are a continuation of the studies described in the previous paper [13]. We wish to report the more profound evaluation of potential herbicidal efficiency of these thiophene-based aminophosphonates **2a**–**h**. Their phytotoxicity was tested on three popular compounds in Poland and persistent weeds: gallant soldier (*Galinsoga parviflora* Cav.), common sorrel (*Rumex acetosa* L.), and white goosefoot (*Chenopodium album* L.). All three plants are dicotyledons. Following recommendations by the European Chemicals Agency (ECHA), we evaluated ecotoxicological impact of the tested compounds on crustaceans *Heterocypris incongruens* (Ostaracodtoxkit™) and bacteria *Aliivibrio fischeri* (Microtox^®®^).

## 2. Results

### 2.1. Synthesis of Aminophosphonates ***2a***–***h***

Previously published aminophosphonates **2a**–**c** and **2e**–**g** were synthesized following the reported procedure [13] by the aza-Pudovik reaction i.e., by the addition of dimethyl phosphite to the azomethine bond of thiophene-2-carbaldimines **1a**–**h**. However, some significant modifications were introduced to this method, i.e., preparation of aldimines **1a**–**c** and **1e**–**g** was carried out in dry methanol in the presence of molecular sieves 3A for 48 h at room temperature. Thus, obtained aldimines **1a**–**h** were used for a phosphite addition without being isolated since it was described previously [13]. (Scheme 1) Preparation of aminophosphonates **2a**–**c** and **2e**–**h** was carried out in boiling acetonitrile following exactly the procedure described in reference [13] with an exception of the reaction time, which was prolonged to seven days.

Dimethyl *N*-(*tert*-butyl)(2-thienyl)methylphosphonate (**2h**) was prepared using the above method and was obtained in 66% yield after purification by column chromatography.

Alternatively, dimethyl N-(2-methoxyphenyl)amino(2-thienyl)methylphosphonate (**2d**) was prepared by stirring an amine with thiophene-2-carboxaldehyde in dry toluene in the presence of magnesium sulfate for 3 h, which was followed by the addition of dimethyl phosphite. The reaction was then continued in toluene at 100 °C for 8 h to obtain 2d in 57% yield after purification by column chromatography.

Aminophosphonates **2a**–**c** and **2e**–**g** were described in our previous paper [13] and, after the workup described in Section 3, they were identified by comparison to authentic samples and were found identical. Aminophosphonates **2d** and **2h** are new compounds and have never been described in the literature. Therefore, they were characterized and identified by ^1^H, ^13^C, and ^31^P NMR spectroscopy and their purity was verified by means of the elemental analysis. Their spectral analysis is described in Section 3 and scans of their NMR spectra are collected in Appendix A.

### 2.2. Phytotoxicity of Tested Aminophosphonates ***2a***–***h***

The plant growth test according to the OECD 208 Standard [14] is a representative, reliable, and well-accepted method to determine the phytotoxicological impact of chemical substances. This method was already successfully used by authors to evaluate the toxicity effect of such various aminophosphonic derivatives [13,15,16,17].

In this study, the efficacy of herbicidal activity of selected aminophosphonates **2a**–**h** against three species of dicotyledonous persistent weeds known as gallant soldier (*Galinsoga parviflora* Cav.), common sorrel (*Rumex acetosa* L.), and white goosefoot (*Chenopodium album* L.), has been evaluated. Digital photographs of weeds one day before determination of shoot inhibition are presented in Figure 1. Their analysis showed that the substances **2a**, **2g**, and **2h** are the most toxic to all weeds. For comparison, substances **2d** and **2e** have been found as the almost nontoxic compounds.

For aminophosphonates bearing a methylphenyl group, dimethyl *N*-(2-methylphenyl)amino- (2-thienyl)methylphosphonate (**2a**) was discovered to be the most toxic to all tested weeds. Toxic effects of **2a** was observed at the highest concentration used (1000 mg/kg s.d.w.), which resulted in total inhibition of germination and plant growth of all tested plants: *Galinsoga parviflora* Cav., *Chenopodium album* L. and *Rumex acetosa* L. Less toxicity to all weeds has been determined for a thiophene derivative bearing a *m*-methylphenyl moiety (**2b**) while the lowest for a para substituted compound (**2c**).

A different tendency was demonstrated for aminophosphonates bearing a methoxyphenyl moiety (**2d**–**f**). Among the *N*-methoxyphenyl-substituted aminophosphonates, a *p*-methyl-substituted derivative (**2f**) was found to be the most toxic among all methoxy-substituted compounds (**2d**–**f**). The susceptibility degree of plants to the derivative **2f** was in the following order: *Rumex acetosa* L. > *Galinsoga parviflora* cav. > *Chenopodium album* L. (Table 1). Comparing to methylphenyl-substituted compounds **2a**–**c**, methoxyphenyl-substituted derivatives **2d**–**f** were less toxic for all weeds used in the test.

Phytotoxicity of aminophosphonates **2g** and **2h** is comparable. However, dimethyl *N*-benzylamino(2-thienyl)methylphosphonate (**2g**) was slightly more toxic against the gallant soldier and white goosefoot when compared to dimethyl *N*-(*t*-butyl)amino(2-thienyl)- methylphosphonate (**2h**). At the highest concentration the negative impact of **2g** on shoot growth was as follows: 100% GI (*Rumex acetosa* L.) > 97% GI (*Chenopodium album* L.) > 95% GI (*Galinsoga parviflora* Cav.) and for **2h** 84%GI (*Rumex acetosa* L.) > 93% GI (*Chenopodium album* L.) > 92% GI (*Galinsoga parviflora* Cav.).

Herbicides applied in soil penetrate the plant through the seed coat. The difference in efficiency may result from their different morphological structure, which determines the retention and penetration of the agent. It is worth noticing that *Chenopodium album* L. showed the lowest sensitivity on all tested substances. This weed is a broad-leaved, annual, and ruderal herb, which is listed as the world′s 10th most serious weed found in Europe. It is able to adapt and to grow vigorously in many different climates and soils [18]. *Chenopodium album* is considered to be one of the super weeds (herbicide-resistant weeds), that is why the action of the tested substances was not as effective as on the other weeds.

Changes of effective concentration EC_50_ values (expressed in mg per kg of soil dry weight) are presented in Figure 2. Comparing *N*-methylphenyl-substituted aminophosphonates **2a**–**c**, it was found that aminophosphonate bearing an *o*-methylphenyl moiety (**2a**) has the most inhibiting influence on the plant growth of all weeds when compared to **2b** and **2c** (*Galinsoga parviflora* Cav. EC_50_ = 275.6 mg/kg; *Chenopodium album* L. EC_50_ = 434 mg/kg, *Rumex acetosa* L. EC_50_ = 186.7 mg/kg). *N*-(3-methylphenyl) aminophosphonate (**2b**) was less toxic than **2a**. The values of effective concentration EC_50_ for this compound was in the following order: *Chenopodium album* L. EC_50_ = 572.3 mg/kg > *Galinsoga parviflora* Cav. equalled EC_50_ = 558.2 mg/kg > *Rumex acetosa* L. EC_50_ = 212.2 mg/kg. The lowest toxicity was found for dimethyl *N*-(4-methylphenyl)amino(2-thienyl)methylphosphonate (**2c**) and its EC_50_ value for *Chenopodium album* L. was 1496 mg/kg. For comparison, the EC_50_ value was ca. twice lower for gallant soldier 775.8 mg/kg and common sorrel 687.4 mg/kg. Aminophosphonates with a metoxyphenyl moiety **2d**–**f** are less toxic as compared to **2a**–**c** and their toxic impact is as follows: **2f** > **2e** > **2d**. *N*-(4-Metoxyphenyl) aminophosphonate **2f** showed the lowest toxicity to *Chenopodium album* L. (EC_50_ = 2908 mg/kg), but its toxicity was more than two and three times greater for *Galinsoga parviflora* Cav. and *Rumex acetosa* L. (EC_50_= 1240 mg/kg and EC_50_ = 950 mg/kg, respectively). Compounds substituted in ortho and meta positions (**2d** and **2e**, respectively) were slightly less toxic than **2f**. These substances were found to be almost nontoxic against *Galinsoga parviflora* Cav. and *Chenopodium album* L. (EC_50_ > 3500 mg/kg). The values of effective concentration EC_50_ for aminophosphonates **2g** and **2h** were the lowest when compared to five other tested compounds **2b**–**f** and comparable with values for **2a**. The exact values were as follows: for *Galinsoga parviflora* Cav. EC_50_ = 227 mg/kg, *Chenopodium album* L. EC_50_ = 452.3 mg/kg, *Rumex acetosa* L. EC_50_ = 224.9 mg/kg (for **2g**), and *Galinsoga parviflora* Cav. EC_50_ = 248.1 mg/kg; *Chenopodium album* L. EC_50_ = 479.6 mg/kg, *Rumex acetosa* L. EC_50_ = 240.4 mg/kg (for **2h**).

The visual impact of tested substances **2a**–**h** on examined species of weeds is presented in Figure 3. Ratings were assigned based on scales from the European Weed Research Council (EWRC) and are presented in Table 2. The aminophosphonate **2a** showed the highest herbicidal efficacy on all weeds and, according to the EWRC scale, is ranked as 1: total plant death (100%). The total mortal effect against common sorrel was also demonstrated by the aminophosphonate **2g**. In the case of gallant soldier and white goosefoot, dimethyl *N*-benzylamino(2-thienyl)- methylphosphonate (**2g**) was revealed as 3: very good (95.0% to 97.9%), according to the EWRC scale [19]. Good to acceptable (4) results were observed for the substance **2h** for *Galinsoga parviflora* Cav. and *Chenopodium album* L. (90% to 94.9%). However, its effect on *Rumex acetosa* L. is 5: moderate (82.0% to 89.9%). In addition, the poisoning of *Rumex acetosa* L. by aminophosphonate **2b** was good to acceptable (4). The aminophosphonates **2c**–**f** were found to be substances ineffective as herbicides (EWRC: 7: bad, 55.0% to 69.9%, 8: very bad, 30% to 54.9% and 9: none 0.0% to 29.9%).

### 2.3. Evaluation of Ecotoxicity of Tested Compounds ***2a***–***h***

#### 2.3.1. Microtox^®^ Assay with *Aliivibrio fischeri*

In order to verify the usefulness of tested aminophosphonates as herbicides, it was necessary to evaluate their ecotoxicity. According to the recommendation by the European Chemicals Agency (ECHA) expressed in the REACH (Registration, Evaluation, and Authorisation of Chemicals) procedure, it is recommended to evaluate ecotoxicity of each newly synthesized compound of potential biocidal application. The luminescent, marine bacteria *Aliivibrio fischeri* test is one of various commonly used bio-tests. Values of EC_50_ calculated by Microtox Analyzer software are plotted in Figure 4 and collected in Table 3. The results of toxicity assessment using *Aliivibrio fischeri* as examined organisms showed different toxicity of tested compounds **2a**–**h**.

Analyzing these data, it is worth noting that none of the tested compounds can be considered toxic for *Aliivibrio fischeri*. The most harmful compound is **2g** (EC_50_ = 222.6 mg/L). Aminophosphonates **2e** and **2f** were even less harmful with EC_50_ values of 308.4 mg/L and 349.9 mg/L, respectively. Organisms used in the test showed higher resistance to compounds **2b** and **2d** (EC_50_ = 399.4 mg/L and 420 mg/L, respectively). The EC_50_ value for aminophosphonate **2h** was 747.5 mg/L while the derivative **2a** was found to be the most harmless with an EC_50_ value 1345 mg/L.

From among three above-mentioned aminophosphonates of herbicidal potential, only two were found to be harmless for *Aliivibrio fischeri* at concentrations giving a half-maximal response (EC_50_). For the aminophosphonate **2a** (*Galinsoga parviflora* Cav. EC_50_ = 275.6 mg/kg, *Chenopodium album* L. EC_50_ = 434 mg/kg, *Rumex acetosa* L. EC_50_= 186.7 mg/kg), the EC_50_ value for *Aliivibrio fischeri* is 1775.4 mg/kg s.d.w. The aminophosphonate **2h** (*Galinsoga parviflora* Cav. EC_50_ = 248.1 mg/kg, *Chenopodium album* L. EC_50_ = 479.6 mg/kg, *Rumex acetosa* L. EC_50_ = 240.4 mg/kg) gave the EC_50_ value of 986.7 mg/kg s.d.w. for *Aliivibrio fischeri*. In order to facilitate the comparison, EC_50_ values for *Aliivibrio fischeri* were recalculated in mg/kg of soil dry weight (see Table 3).

It is worth noticing that, in the case of aminophosphonates, **2a**–**c** bearing a methylphenyl group are much less harmful for *Aliivibrio fischeri* than derivatives **2d**–**f** having a methoxyphenyl moiety. From among compounds **2a**–**f**, the highest harmful impact was observed for molecules with meta substitution (**2b** and **2e**) while aminophosphonates have a phenyl ring substituted in ortho positions characterized with the lowest harmfulness to bacteria *Aliivibrio fischeri*.

According to Hernando et al. [20], the toxicity categories against *Aliivibrio fischeri* based on the EC_50_ values are: “very toxic to aquatic organisms” (EC_50_ ≤ 1 mg/L), “toxic” (EC_50_ in the range of 1–10 mg/L), and “harmful” (EC_50_ in the range of 10–100 mg/L), which are established in the Directive 93/67/EEC [21].

Therefore, all studied aminophosphonates **2a**–**h** have EC_50_ high above 100 mg/L, which may be classified as nearly harmless.

#### 2.3.2. OSTRACODTOXKIT™ Test with *Heterocypris incongruens*

Evaluation of aminophosphonates **2a**–**h** toxicity against *Heterocypris incongruens* revealed that all tested compounds are very toxic for this species. Aminophosphonates **2b**–**h** caused over 50% mortality of ostracods at the lowest concentration (10 mg/kg s.d.w.) while dimethyl *N*-(2-methylphenyl)amino(2-thienyl)methylphosphonate (**2a**) caused death of 48% of tested population tendency (Figure 5). It is to stress that the total mortality of *Heterocypris incrogruens* has been found observed at the concentration 50 mg/kg s.d.w. of thiophene derivatives **2a**–**h**. According to the Ostracodtoxkit™ test manual, growth inhibition (the second parameter of test) should only be determined for sediments with mortality that is less than 30% and that is why this parameter of the Ostracod Test was not measured.

The most toxic substance at the lowest concentration 10 mg/kg was dimethyl *N*-benzylamino(2-thienyl)methylphosphonate **2g** and toxicity of all the rest was in order as follows: **2e** > **2f** > **2b** > **2d** > **2c** > **2h** > **2a**.

Among *N*-methylphenyl (**2a**–**c**) and *N*-methoxyphenyl aminophosphonates **2d**–**f**, derivatives having a meta substitution (**2b** and **2e**) were found to be the most toxic. Slightly lower toxicity was observed for the thiophene derivatives with *p*-methylphenyl (**2c**) and *p*-methoxyphenyl (**2f**) moieties and the least toxicity was found for the ortho-substituted derivatives. As it was observed for toxicity against *Aliivibrio fischeri*, *N*-methoxyphenyl aminophosphonates **2d**–**f** were definitely more toxic than *N*-methylphenyl derivatives (**2a**–**c**).

In this scenario, the investigated thiophene-derived aminophosphonates **2a**–**h** showed much greater toxicity to crustaceans than pyrrole-, 5-nitrofurfuryl-, and ferrocene-derived aminophosphonates, which we reported previously [15,16,17].

## 3. Discussion

In our previous paper [13], we reported the phytotoxicological study on a series of amino(2-thienyl)-methylphosphonates where their impact on common radish (*Raphanus sativus*) and oat (*Avena sativa*) was measured and characterized by a NOEC/LOEC factor and by the half maximal effective concentration (EC_50_) [13]. In this scenario, amino(2-thienyl)methylphosphonates (**2a**–**h**) of this series were investigated in aspect of their impact on three common and persistent weeds: gallant soldier (*Galinsoga parviflora* Cav.), white goosefoot (*Chenopodium album* L.), and common sorrel (*Rumex acetosa* L.). These studies aimed at selecting candidates for potential new herbicides.

Analyzing collected data (Figure 1, Figure 2 and Figure 3, Table 2) in the point of view of an herbicidal potency, one notes that compounds **2c**–**f** are totally out of interest. The EC_50_ values of growth inhibition of tested plant shoots following their exposure are indeed high (775.8–3667 mg/kg s.d.w. for gallant soldier, 1496–3789 mg/kg s.d.w. for white goosefoot and 687.4–1333 mg/kg s.d.w. for common sorrel). Their scores according to the EWRC rating scale were estimated 7–9 [19]. These results coincide well with previously published [13] phytotoxicity data for oat and radish since these compounds were found harmless for both plants (EC_50_ in the range 553.4–1192 mg/kg s.d.w. for radish and 1241–1523 mg/kg s.d.w. for oat). Moreover, they were found to be highly toxic against freshwater crustaceans *Heterocypris incongruens* since the compounds **2c**–**f** caused their mortality ranging from 59% to 83% at the concentration 10 mg/kg s.d.w., which must be kept in mind at any potential application of these compounds.

Much better results were obtained for aminophosphonates **2b** and **2h**. The EC_50_ values of growth inhibition of tested plant shoots following their exposure were much lower, which indicates the higher phytotoxicity (for **2b**: 558.2 mg/kg s.d.w. for gallant soldier, 572.3 mg/kg s.d.w. for white goosefoot, and 212.2 mg/kg s.d.w. for common sorrel. For **2h**: 248.1 mg/kg s.d.w. for gallant soldier, 479.6 mg/kg s.d.w. for white goosefoot, and 240.4 mg/kg s.d.w. for common sorrel). Their EWRC toxicity scale ranks were estimated as follows: for **2b**: 6 for gallant soldier, 6 for white goosefoot, and 4 for the common sorrel. For **2h**: 4 for gallant soldier, 4 for white goosefoot, and 5 for common sorrel. Both compounds **2b** and **2h** are also toxic for crustaceans *Heterocypris incongruens* (72% and 55%, respectively, Figure 5). These data, despite being better than results obtained for aminophosphonates **2c**–**f**, show clearly that compounds **2b** and **2h** cannot be considered as potential herbicides.

The best results were obtained for dimethyl *N*-(2-methylphenyl)amino(2-thienyl)methyl- phosphonate (**2a**) and dimethyl *N*-benzylamino(2-thienyl)methylphosphonate (**2g**). Benzyl derivative **2g** was found to be selectively toxic against radish (*R. sativus*) [13] (EC_50_ = 379.7 mg/kg s.d.w. vs. 863.4 mg/kg s.d.w. for oat). Its toxicity against weeds may be considered as promising (EC_50_ values of growth inhibition were 227 mg/kg s.d.w. for *Galinsoga parviflora* Cav., 452.3 mg/kg s.d.w. for *Chenopodium album* L., and 224.9 mg/kg s.d.w. for *Rumex acetosa* L. and its EWRC toxicity scale ranks were 3, 3, and 1, respectively, Table 2). However, its toxicity against *Heterocypris incongruens* is the highest from among all tested compounds (87% at 10 mg/kg s.d.w., Figure 5).

Dimethyl *N*-(2-methylphenyl)amino(2-thienyl)methylphosphonate (**2a**) was found to have the best phytotoxicological properties. Its toxicity against tested weeds was high. EC_50_ values of growth inhibition were 275.6 mg/kg s.d.w. for *Galinsoga parviflora* Cav., 434 mg/kg s.d.w. for *Chenopodium album* L., and 186.7 mg/kg s.d.w. for *Rumex acetosa* L. and its scores, according to the EWRC rating scale [19], which were 1 for each plant and which signifies total plant death (100%) at 1000 mg/kg s.d.w. Its toxicity against radish [13] was found to be highly selective and EC_50_ was 355.6 mg/kg s.d.w. vs. EC_50_ = 1284 mg/kg s.d.w. and NOEC = 400 mg/kg s.d.w. for oat. Especially the last value for oat, i.e., the no observed effect concentration (NOEC) is significant, as values of the half maximal effective concentration for gallant soldier (*Galinsoga parviflora* Cav.) and common sorrel (*Rumex acetosa* L.) are below 400 mg/kg. It means that, at concentration, which is necessary for important intoxication of both weeds, oat is safe (Figure 1 and Figure 2). Important damage of white goosefoot (*Chenopodium album* L.) is observed at 400 mg/kg s.d.w. of **2a** (Figure 1), which concentration is still harmless for oat.

Despite all tested compounds, **2a**–**h** were found to be harmless for luminescent marine bacteria *Aliivibrio fischeri*. In fact, the aminophosphonate **2a** is the most safe for these bacteria (EC_50_ = 1775.4 mg/kg s.d.w.).

The most popular and commonly used herbicides are not free of various flaws. Alshallash [22] reported that the length of oat, sorghum, and sugar beet roots was reduced after the treatment with pendimethalin. Moreover, the herbicide was also phytotoxic to soybeans. When pendimethalin and trifluralin were used to control ryegrass (*Lolium multiflorum*) associated with barley (*Hordeum vulgare*) [22], a strong phytotoxicological effect on barley was observed. Both herbicides controlled well growth of *Lolium multiflorum* and the effect of pendimethalin on barley was very negative. Trifluralin was found much less harmful to barley plants. The toxic impact of pendimethalin was confirmed for soybean and cowpea [23], but residues of pre-plant pendimethalin were found not to affect the oat crop [24].

Sensitivity of spring planted cereals (barley, oats, and wheat) to mesotrione was examined [25] and it was proved that mesotrione applied pre-emergence (PRE) caused minimal visible injury and had no adverse effect on plant height or yield. However, post-emergence (POST) application of mesotrione resulted in important injury and reduced plant height. Regarding these results, mesotrione should be applied in reduced doses to control weeds [26], which was reported to be effective since such a procedure gave good results depending on the spectrum of broadleaved weed species [26,27].

Glyphosate, which is the most popular and most controversial herbicide, is certainly non-selective. The harmful impact of the glyphosate-based formulation on the red radish and barley was confirmed [28] and it was revealed that barley was slightly more sensitive to Roundup formulations than red radish [28]. Our tests performed with glyphosate [29] in the form of acid (not salt) showed that it was nearly equally harmful for oat and radish shoots and caused slightly stronger effects on radish roots and fresh matter. However, NOEC/LOEC values were the same for both plants (NOEC = 100 and LOEC = 200 mg/kg s.d.w.). Its impact on gallant soldier (*Galinsoga parviflora* Cav.), common sorrel (*Rumex acetosa* L.), and white goosefoot (*Chenopodium album* L.) was evaluated, according to the EWRC toxicity scale [18] and was found to be 5, 5, and 9, respectively, which demonstrates that an acid form of glyphosate shows a moderate herbicidal activity. Its pesticide formulations such as Roundup^®^ are undoubtedly more efficient due to the added adjuvants (surfactants) and the lack of these adjuvants in the herbicidal formulation dramatically decreases its efficiency, which is mostly caused by the loss of an active agent not having enough adhesive properties [30]. However, the problem is that pesticide formulations such as Roundup^®^ are characterized by the toxicity 17–32 times higher than glyphosate [9].

Several of commonly used herbicides have an adverse and harmful impact on luminescent bacteria *Aliivibrio fischeri*, which are commonly applied in tests evaluating ecotoxicological properties of chemical compounds. Mesotrione neutralized with a phosphate buffer showed toxicity against *Aliivibrio fischeri* with IC_50_ = 69.2 mg/L [31,32] while, in a non-neutralized form, its IC_50_ = 43.6 mg/L [32]. Pendimethalin, after 15 min of contact with *Aliivibrio fischeri*, showed toxicity with IC_50_ = 225.0 mg/L [33,34], which is not elevated but much higher than a value for the compound **2a**.

Toxicity of glyphosate against *Aliivibrio fischeri* was measured several times in various conditions. The measurements carried out on a Microtox^®^ system [35] demonstrated its EC_50_ = 18.23 mg/L after 15 min of contact while tests performed for a glyphosate metabolite–aminomethylphosphonic acid showed toxicity with EC50 = 53.43 mg/L [35]. The Microtox^®®^ test on *Aliivibrio fischeri* with glyphosate by Tsui and Chu [36] gave a similar value (EC_50_ = 17.5 mg/L). Tests performed with *Aliivibrio fischeri* on a BioFix^®®^ Lumi system gave EC_50_ value to be 43.8 mg/L after 15 min [20].

It is to note that the toxic impact of dimethyl *N*-(2-methylphenyl)amino(2-thienyl)methyl- phosphonate (**2a**) on ostracods (*Heterocypris incongruens*) was detected to be the lowest among all tested compounds **2a**–**h** and it killed barely 48% of the tested population of ostracods at 10 mg/kg s.d.w. Certainly, we realize that it is still a significant toxicity, which call into question the possible application of this compound as a herbicide.

Pendimethalin, which is a largely and commonly used herbicide approved in a majority of European countries in the USA and in Australia [37], was classified to be moderately toxic against crustaceans *Daphnia magna*, according to the IUPAC database [37]. According to other sources, it is considered to be highly toxic [38,39,40]. Among other herbicides, which are approved in various countries, diquat, glyphosate, and linuron are classified as moderately toxic to *D. magna* and oxyfluorfen, is claimed to be highly toxic. The U.S. Environmental Protection Agency (EPA) published [38] the qualitative descriptors for categories of fish and aquatic invertebrate toxicity where it is stated that the toxicity of a compound (or preparation) expressed as LC_50_ or EC_50_ is below 0.1 ppm. It is classified as very highly toxic. When the value is between 0.1–1 ppm, a compound is considered to be highly toxic and, when it is within the range 1–10 ppm, it is considered to be moderately toxic.

The toxic impact of the aminophosphonate **2a** caused the 48% mortality of crustaceans *Heterocypris incongruens* at the concentration 10 mg/kg s.d.w., i.e., 7.58 mg/L. Therefore, undoubtedly, the LC_50_ value of the compound **2a** lied within the range 1–10 ppm and the substance **2a** should be classified as moderately toxic. Due to this reason, the investigated and selected compound **2a** can be considered as the potential herbicide and its toxicological properties are to be evaluated extensively and profoundly. It is, however, to admit that toxicity of **2a** against freshwater crustaceans *Heterocypris incongruens* could be problematic with regard to its potential herbicidal application. The reasonable solution for this problem is the controlled release of a herbicide from a biodegradable polymer carrier where small amounts of an active ingredient are liberated. This process will be the subject of our investigation in the nearest future.

## 4. Materials and Methods

All solvents (POCh, Gliwice, Poland) were routinely distilled and dried prior to use. Amines, diphenyl phosphite, and pyrrole-2-carboxaldehyde (Aldrich, Poznań, Poland) were used as received. Melting points were measured on a MelTemp II apparatus (Bibby Scientific Limited, Staffordshire, UK) in a capillary and were not corrected. NMR spectra were recorded on an Avance III 600 MHz apparatus (Bruker, Billerica, MA, USA) operating at 600 MHz (^1^H-NMR), 150 MHz (^13^C-NMR), and 243 MHz (^31^P-NMR). Elemental analyses were carried out at the Laboratory of Microanalysis, Faculty of Chemistry, University of Łódź, Poland.

### 4.1. Chemistry

#### 4.1.1. Preparation of Amino(2-thienyl)methylphosphonates **2a**–**c** and **2e**–**h**

To a solution of thiophene-2-carboxaldehyde (4.48 g, 40 mmol) in dry methanol (50 mL) with molecular sieves 3A, a solution of an appropriate amine (40 mmol) in dry methanol (20 mL) was added. The obtained mixture was stirred for seven days at room temperature. Afterward, the sieves were filtered off and the filtrate was evaporated to achieve crude imines, which were used for the further reaction without any purification.

Crude imine (40 mmol) obtained in a previously described procedure was dissolved in acetonitrile (50 mL) and dimethyl phosphite (8.8 g, 80 mmol) was added. The mixture was then refluxed with stirring for seven days. Then, the solvent and all volatile components were removed *in vacuo*. Residue was dissolved in DCM (100 mL) and a solution was washed with saturated aqueous sodium hydrogen carbonate (4 × 25 mL). Collected organic layers were then dried over MgSO_4_. Then the drying agent was filtered off, DCM was evaporated, and the products **2a**–**c** and **2e**–**g** were purified by flash column chromatography, as it was described [13]. They were identified by comparison to authentic samples [13] and were found identical. Their spectra can be found in our previous paper published in Molecules [13].

*Dimethyl N-(tert-butyl)(2-thienyl)methylphosphonate* (**2h**): Separated by double flash chromatography: 1^st^: DCM:AcOEt = 1:1, R_f_ = 0,36; 2^nd^: petroleum ether:AcOEt = 1:1, R_f_ = 0.16. Yield = 66% (7.31 g) as light yellow crystals, mp = 48–49 °C. ^1^H NMR (CDCl_3_, 600 MHz): δ 7.22–7.21 (m, H_5_^thioph^, 1H), 7.09–7.08 (m, H_3_^thioph^, 1H), 6.97–6.96 (m, H_4_^thioph^, 1H), 5.10 (d, ^2^J_PH_ = 24.0 Hz, CHP, 1H), 3.79 (d, ^3^J_PH_ = 10.8 Hz, POCH_3_, 3H), 3.59 (d, ^3^J_PH_ = 10.8 Hz, POCH_3_, 3H); 1.06 (s, CCH_3_, 9H). ^13^C NMR (CDCl_3_, 150 MHz): δ 143.82 (C^2^_thioph_), 126.93 (d, ^4^J_CP_ = 2.6 Hz, C^5^_thioph_), 125.4 (d, ^2^J_CP_ = 7.8 Hz, C^3^_thioph_), 124.8 (d, ^3^J_CP_ = 4.0 Hz, C^4^_thioph_), 54.50 (d, ^2^J_CP_ = 6.7 Hz, PO*C*), 53.37 (d, ^2^J_CP_ = 6.7 Hz, PO*C*), 52.43 (d, ^3^J_CP_ = 14.0 Hz, CH_3_*C*), 50.76 (d, ^1^J_CP_ = 161.2 Hz, P*C*), 29.63 (CH_3_).^31^P NMR (243 MHz, CDCl_3_): δ 24.60. *Elem. Anal.* Calcd for C_11_H_20_NO_3_PS: C, 47.64, H, 7.27, N, 5.05. Found: C, 47.71, H, 7.25, N, 4.99.

#### 4.1.2. Preparation of Dimethyl N-(2-methoxyphenyl)amino(2-thienyl)methylphosphonate (**2d**)

To a solution of thiophene-2-carboxaldehyde (4.48 g, 40 mmol) in dry toluene (50 mL) with magnesium sulfate (4.8 g, 40 mmol) and an appropriate amine (40 mmol) was added in several portions. A mixture was stirred at room temperature for 3 h. Then, dimethyl phosphite (8.8 g, 80 mmol) was added and a whole mixture was heated at 100 °C for 8 h. Then 50 mL of DCM was added and a mixture was concentrated in vacuo. Residue was dissolved in DCM (100 mL) and a solution was washed with saturated aqueous sodium hydrogen carbonate (4 × 25 mL). Collected organic layers were then dried over MgSO_4_. Then the drying agent was filtered off, DCM was evaporated, and the product was purified by double flash column chromatography on silica gel: 1^st^: DCM:AcOEt = 1:1 R_f_ = 0,5, 2^nd^: AcOEt:hexane = 5:1, R_f_ = 0.44.

Yield = 57% (7.46 g) as light yellow crystals, mp = 68–69 °C. ^1^H NMR (CDCl_3_, 600 MHz): δ 7.21 (ddd, ^3^J_HH_ = 4.8 Hz, ^4^J_HH_ = 1.8 Hz, ^4^J_PH_ = 1.2 Hz, H_3_^thioph^, 1H), 7.17–7.16 (m, H_5_^thioph^, 1H), 6.96 (dd, ^3^J_HH_ = 4.8 Hz and 3.6 Hz, H_4_^thioph^, 1H), 6.79–6.76 (m, *o*-C_6_H_4_, 2H), 6.72–6.69 (m, *o*-C_6_H_4_, 1H), 6.60–6.59 (m, *o*-C_6_H_4_, 1H), 5.17 (dd, ^3^J_HH_ = 8.4 Hz and ^3^J_PH_ = 7.8 Hz, NH, 1H), 5.07 (dd, ^3^J_HH_ = 8.4 Hz and ^2^J_PH_ = 23.4 Hz, CHP, 1H), 3.86 (s, OCH_3_, 3H), 3.78 (d, ^3^J_PH_ = 10.8 Hz, POCH_3_, 3H), 3.65 (d, ^3^J_PH_ = 10.2 Hz, POCH_3_, 3H). ^13^C NMR (CDCl_3_, 150 MHz): δ 147.5 (C_arom_), 139.7 (d, ^2^J_CP_ = 1.4 Hz, C_arom_), 135.8 (d, ^2^J_CP_ = 13.1 Hz, C^2^_thioph_), 127.1 (d, ^4^J_CP_ = 2.9 Hz, C_thioph_), 126.2 (d, ^2^J_CP_ = 7.2 Hz, C_thioph_), 125.4 (d, ^3^J_CP_ = 3.6 Hz, C_thioph_), 121.0 (C_arom_), 118.4 (C_arom_), 111.3 (C_arom_), 109.9 (C_arom_), 55.6 (ArO*C*), 54.2 (d, ^2^J_CP_ = 7.0 Hz, PO*C*), 53.8 (d, ^2^J_CP_ = 7.0 Hz, PO*C*), and 51.5 (d, ^1^J_CP_ = 159.5 Hz, P*C*). ^31^P NMR (243 MHz, CDCl_3_): δ 23.12. *Elem. Anal.* Calcd for C_14_H_18_NO_4_PS: C, 51.37, H, 5.54, N, 4.28. Found: C, 51.41, H, 5.48, N, 4.30.

### 4.2. Plant Growth Test of Aminophosphonates ***2a**–**h***

The plant growth test for aminophosphonates was carried out in laboratory conditions following the OECD 208 Guideline Terrestrial Plants Growth Test [14] for the gallant soldier (*Galinsoga parviflora* Cav.), common sorrel (*Rumex acetosa* L.), and white goosefoot (*Chenopodium album L*.)

According to the OECD 208 standard [14], the plant growth test of compounds **2a**–**h** was carried out in sandy soil with the following parameters: granulometric composition of soil 77% sand, 16% dust, and loam, organic carbon content of approximately 1.6%, pH (KCl) equal to 6.6.

The tests were carried out in polypropylene pots (diameter of 90 mm and capacity of 300 cm^3^), which were filled with the control soil or with the soil mixed with the tested compounds added at the following concentrations: 100, 400, and 1000 mg/kg of soil dry weight (s.d.w.). Each concentration was done in triplicate (three pots for gallant soldier, common sorrel, and white goosefoot). The seeds of each of the selected plant species were sown into the soil. Seeds originated from the same source. Plants were grown for six weeks under controlled conditions: a constant humidity content at the level required for the plants (70% field water capacity), temperature (20 ± 2 °C), and light intensity (7000lux) in the system of 16 h/day and 8 h/night.

The evaluation of phytotoxicity of the studied aminophosphonates **2a**–**h** at applied concentrations was made by comparing the determined growth inhibition of the shoot of selected plants were measured, as described previously by References [15,16,17]. The length of plants is defined as the length of the tip of the longest leaf to the base of culms. The inhibition ratio (IR) was calculated, according to the method given by Lewkowski et al. [15,16,17], namely:
$$IR\%=\left(1-\frac{length\;in\;treatment\;group}{length\;in\;control\;group}\right)\times 100\%$$

The effective concentration EC_50_ for the shoot of plants was calculated using GraphPad Prism software (Version 7, GraphPad Software, Inc., La Jolla, CA, USA).

Herbicidal activity of tested compounds **2a**–**h** was determined by visual assessment of growth inhibition, damages, and their drying out and was documented on digital photographs, which was made on the sixth week of plant growth. Scores were assigned according to the European Weed Research Council (EWRC) rating scale [19,41] for weed control: 1: total plant death (100%), 2: excellent (98.0% to 99.9%), 3: very good (95.0% to 97.9%), 4: good to acceptable (90% to 94.9%), 5: moderate (82.0% to 89.9%), 6: weak (70.0% to 81.9%), 7: bad (55.0% to 69.9%), 8: very bad (30% to 54.9%), and 9: none (0.0% to 29.9%).

### 4.3. Microtox^®®^ Toxicity Assay

A detailed procedure of the Microtox Toxicity Assay has been described previously by our group [42]. The method is based on the analysis of light emission reduction of luminescent bacteria (*Aliivibrio fischeri*) under toxic stress. The tests were carried out in a Microtox^®®^ M500 analyzer, according to the 1992 Microtox^®®^ Manual. The Microtox^®®^ Solid-Phase Test (MSPT) was performed following the report by Doe et al. [43].

The MSPT procedure allows the test organisms to have direct contact with the solid sample in an aqueous suspension of the test sample. Therefore, it is possible to detect toxicity, which is caused by the insoluble solids and which are not in the solution. All materials and reagents were purchased from MODERNWATER (New Castle, DE, USA). To determine toxicity, the marine luminescent bacterium, *Aliivibrio fischeri*, were used since they are naturally adapted to a saline environment. The bacteria were regenerated with 1 mL of Reconstitution Solution (0.01%) and placed in the reagent well of the Microtox^®®^. A suspension of 7 g of the tested soil was prepared in 35 mL of a Solid Phase Diluent (3.5% NaCl) and was stirred on a magnetic stirrer for 10 min. Then a series of dilutions were made and bacteria (approximately 1 × 10^6^ cell/mL per assay) were exposed to these dilutions and to a blank (3.5% NaCl solution) for 20 min. Then, after filtration, the light output of supernatants containing exposed bacteria was measured after 5 min with a Microtox^®®^ Analyzer 500. The inhibition was calculated as the concentration of the compound loaded to sediment (mg/L) that caused a 50% reduction in the light emitted by the bacteria and EC_50_ along with a 95% confidence limit determined by the software provided with the analyzer.

### 4.4. Ostracod Test Kit

The evaluation of eco-toxicity of synthesized compounds was carried out in a short-term contact test using Ostracodtoxkit F™ provided by MicroBiotests Inc. (Gent, Belgium). This direct sediment contact bioassay was performed in multiwell test plates using neonates of the benthic ostracod crustacean *Heterocypris incongruens* hatched from cysts [44].

After six days of contact with the tested soil, the percentage mortality (calculated the total number of dead ostracods expressed in percent) and the growth of the crustaceans were determined and compared to the results obtained in a non-treated reference soil. According to the manual of Ostracodtoxkit test, the cysts (*Heterocypris incongruens*) were transferred into a Petri dish filled with 10 mL standard fresh water (reconstituted water) and were incubated at 25 °C for 52 h under continuous illumination (approx. 3000–4000 lux). After 48 h of cysts incubation, pre-feeding of the freshly hatched ostracods was performed with algae (spirulina-powder) provided in the test kit. Next, after hatching, before feeding with algal food suspension, the length measurements of ostracod neonates has been done. Algae (*Selenastrum capricornutum*) used as feed in the test plate were reconstituted, according to the manufacturer’s procedure. Each well of a test plate was filled in the following order: 2 mL standard freshwater, 2500 µL of sediment (soil) treated and non-treated for comparison (blank), 2 mL already prepared algal suspension, and 10 ostracods. The test plates were covered with Parafilm^®®^ and closed by a lid. Then multi wall plates were incubated at 25 °C in darkness for six days. After six days of exposure, the ostracods have been recovered from the multiwells to determine the percentage mortality. To calculate the growth inhibition of survived organisms, their length measurements were completed. The mortality of test organisms was determined in six replicates. The measurement of length was carried out by means of a micrometric strip placed on the bottom of a glass microscope plate. Growth inhibition (GI) of *Heterocypris incongruens* in the test sediment was calculated by using the equation (1) below.
(1)$$GI\%=\left(1-\frac{growth\;in\;test\;sed.}{growth\;in\;ref.sed.}\right)\times 100\%$$

## 5. Conclusions

A series of dimethyl aminophosphonates bearing a 2-thienyl moiety was synthesized and synthesized compounds were evaluated in aspect of their phytotoxicological properties. Obtained results showed clearly that aminophosphonates **2c**–**f** do not have any potential as soil applied herbicides due to their very weak inhibitory effect on investigated plants. Dimethyl *N*-(3-methylphenyl)amino(2-thienyl)methylphosphonate (**2b**) and the *N*-(*t*-butyl)amino derivative (**2h**), while slightly more efficient, showed the similar lack of herbicidal potential. Much better herbicidal properties were demonstrated for the *N*-benzylamino derivative (**2g**), but its toxicity against ostracods is the highest among all tested compounds.

Based on the obtained results, dimethyl *N*-(2-methylphenyl)amino(2-thienyl)- methylphosphonate (**2a**) was found as the total weed killer and more comprehensive study of its inhibitory mechanism at the molecular level should be performed.

All investigated compounds **2a**–**h** were also tested in aspect of their ecotoxicological impact on *Heterocypris incongruens* (OSTRACODTOXKIT^TM^) and *Aliivibrio fischeri* (Microtox^®®^ test). The results showed distinctly that all compounds **2a**–**h** are practically not toxic for bacteria *Aliivibrio fischeri*, but they are moderately toxic for crustaceans *Heterocypris incongruens*. It is to note that the aminophosphonate **2a**, which was selected as a potential herbicide showed the lowest toxicity against ostracods. Therefore, the aminophosphonate **2a** has a potential as herbicide and studies on its application as an active ingredient in herbicidal formulation should be performed.

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
