# Peer review of "Effect of New Thiophene-Derived Aminophosphonic Derivatives on Growth of Terrestrial Plants. Part 2. Their Ecotoxicological Impact and Phytotoxicity Test Toward Herbicidal Application in Agriculture"

_molecules, 2018, doi:10.3390/molecules23123173_

Reviewer 1 Report

In this work the effect of new thiophene-derived aminophosphonic derivatives on growth of terrestrial plants is described. Authors evaluated phytotoxicity of synthesized compounds against three persistent weeds of a high degree of resistance. It was found that one of obtained aminophosphonates showed herbicidal potential and it is not toxic against tested bacteria. The article looks like a short communication and may be published after minor revision.

Notes:

1. I think that Introduction should be complemented by the significance of aminophosphonate compounds.

2. The yields of synthesized aminophosphonates should be presented in the Scheme 1.

3. Line 73. In the text is written that synthesis was carried out in boiling acetonitrile. However, in the Scheme 1 toluene is shown. Please, check and correct it.

4. Line 249. The word “Di.scussion” should be corrected and rewritten.

5. Conclusions should be shortened and focused on the main findings of this work. In addition, promising applications of new obtained results should be added.

Author Response

Reviewer’s comments and our responses

Reviewer: In this work the effect of new thiophene-derived aminophosphonic derivatives on growth of terrestrial plants is described. Authors evaluated phytotoxicity of synthesized compounds against three persistent weeds of a high degree of resistance. It was found that one of obtained aminophosphonates showed herbicidal potential and it is not toxic against tested bacteria. The article looks like a short communication and may be published after minor revision.

OUR RESPONSE: Thank you

Notes:

Reviewer: I think that Introduction should be complemented by the significance of aminophosphonate compounds.

OUR RESPONSE: Because the manuscript is a continuation of the paper, which appeared in Molecules (Molecules 2016, 21, 694; doi:10.3390/molecules21060694), where we discussed there the significance of aminophosphonates, we decided not to discuss this topic once again to avoid repeating this text.

Reviewer: The yields of synthesized aminophosphonates should be presented in the Scheme 1.

OUR RESPONSE: We have introduced yields to the scheme

Reviewer: Line 73. In the text is written that synthesis was carried out in boiling acetonitrile. However, in the Scheme 1 toluene is shown. Please, check and correct it.

OUR RESPONSE: Indeed, we have mistaken. The matter is corrected now.

Reviewer: Line 249. The word “Di.scussion” should be corrected and rewritten.

OUR RESPONSE: We have corrected this word (there was no need to rewrite, it was enough to remove a dot:))

Reviewer: Conclusions should be shortened and focused on the main findings of this work. In addition, promising applications of new obtained results should be added.

OUR RESPONSE: The Conclusion part was shortened and we added some words about potential of the compound 2a.

Reviewer 2 Report

The topic falls in the scope of “Molecules” journal. The authors have worked on very important question. Experiments seem well designed and methods used appropriated. Results are well described and conclusions based on the results and do not speculative. However, there are still some issues that have to be addressed by the authors before considering the manuscript for publication. My comments are detailed below.

General comments:

Abstract and throughout the text (including figures): In the spelling of the scientific names of the species, the binomial nomenclature rules should be applied always! Both the first part of the name, the genus, and the second part, the species, should be italicized when a binomial name occurs in normal text, but the botanical authority not.

Similarly, all plant species names should be accompanied by the botanical authority at first mention. In this way, if a plant name is changed in time, it will still be possible to recognize it. For example, in the abstract (lines 16-17) some species names have the botanical authority (Galinsoga parviflora Cav., Rumex acetosa L.) and other species names have not (Chenopodium album). Authors should standardize. “Chenopodium album” should be “Chenopodium album L.”

Specific comments:

 Abstract

I suggest the authors to include some quantitative data which could be more interesting and informative for the readers.

Lines 17-18: “impact on H. incongruens and A. fischeri was measured.” – As is the first mention, the names of the species should be presented in full. Similarly, the species names should be accompanied by the authority name at first mention.

Keywords

Authors should rephrase keywords. Do not use words or terms in the title as keywords: the function of keywords is to supplement the information given in the title. Words in the title are automatically included in indexes, and keywords serve as additional pointers.

Introduction

Line 63: “ECHA” – Abbreviations should be defined at their first mention: “Following recommendations by European Chemicals Agency (ECHA), we evaluated…”

Results

Line 167: “…on scales from European Weed Research Council and are presented in…” should be “…on scales from European Weed Research Council (EWRC) and are presented in…” – Then use the abbreviation throughout the manuscript.

Materials and methods

In this section you should specify the characteristics of all equipments (report model, brand name, city and country of manufacturer).

I cannot understand the statistical analysis (lines 494-497). Why did authors use parametric statistics (ANOVA)? Have the authors check for normality? Authors should explain which test they used for evaluation of the normality of the analysed features and homogeneity of variances. It is known, for the scientist working on evaluation of pollutants in biological matrixes, that these substances never own normal distributions but highly skewed to the left and showing long right tails. Taking this into account I wonder they decided to use directly parametric statistics without (at least this is not noted in the manuscript) any previous evaluation of normality (e.g. Shapiro-Wilk test). For data not showing normal distributions there are a lot of equivalent statistical test that allow to do the same analysis but in a proper way.

Author Response

Reviewer’s comments and our responses

The topic falls in the scope of “Molecules” journal. The authors have worked on very important question. Experiments seem well designed and methods used appropriated. Results are well described and conclusions based on the results and do not speculative. However, there are still some issues that have to be addressed by the authors before considering the manuscript for publication. My comments are detailed below.

OUR RESPONSE: Thank you

Reviewer: Abstract and throughout the text (including figures): In the spelling of the scientific names of the species, the binomial nomenclature rules should be applied always! Both the first part of the name, the genus, and the second part, the species, should be italicized when a binomial name occurs in normal text, but the botanical authority not.

Similarly, all plant species names should be accompanied by the botanical authority at first mention. In this way, if a plant name is changed in time, it will still be possible to recognize it. For example, in the abstract (lines 16-17) some species names have the botanical authority (Galinsoga parviflora Cav., Rumex acetosa L.) and other species names have not (Chenopodium album). Authors should standardize. “Chenopodium album” should be “Chenopodium album L.”

OUR RESPONSE: Thank you for this short overview about botanical nomenclature. The team consists of organic chemists, an environmental engineer and a biotechnologist and no one of us is a true botanist. We have corrected this matter.

Reviewer: Abstract. I suggest the authors to include some quantitative data which could be more interesting and informative for the readers.

OUR RESPONSE: We have introduced some data to the abstract

Reviewer: Lines 17-18: “impact on H. incongruens and A. fischeri was measured.” – As is the first mention, the names of the species should be presented in full. Similarly, the species names should be accompanied by the authority name at first mention.

OUR RESPONSE: We have corrected this matter in a whole manuscript.

Reviewer: Keywords. Authors should rephrase keywords. Do not use words or terms in the title as keywords: the function of keywords is to supplement the information given in the title. Words in the title are automatically included in indexes, and keywords serve as additional pointers.

OUR RESPONSE: We have corrected this matter

Reviewer: Introduction. Line 63: “ECHA” – Abbreviations should be defined at their first mention: “Following recommendations by European Chemicals Agency (ECHA), we evaluated…”

OUR RESPONSE: We did as the referee suggested.

Reviewer: Results. Line 167: “…on scales from European Weed Research Council and are presented in…” should be “…on scales from European Weed Research Council (EWRC) and are presented in…” – Then use the abbreviation throughout the manuscript.

OUR RESPONSE: We have corrected it as the referee suggested.

Reviewer: Materials and methods. In this section you should specify the characteristics of all equipments (report model, brand name, city and country of manufacturer).

OUR RESPONSE: We have introduced these pieces of information

Reviewer: I cannot understand the statistical analysis (lines 494-497). Why did authors use parametric statistics (ANOVA)? Have the authors check for normality? Authors should explain which test they used for evaluation of the normality of the analysed features and homogeneity of variances. It is known, for the scientist working on evaluation of pollutants in biological matrixes, that these substances never own normal distributions but highly skewed to the left and showing long right tails. Taking this into account I wonder they decided to use directly parametric statistics without (at least this is not noted in the manuscript) any previous evaluation of normality (e.g. Shapiro-Wilk test). For data not showing normal distributions there are a lot of equivalent statistical test that allow to do the same analysis but in a proper way.

OUR RESPONSE: Thank you very much for remark related to ANOVA test. Indeed, we agree, that the sentence about ANOVA in section Methods could confuse the reader because we did not use this test for the analysis of obtained results. The mistake is a result of continuation of results description on the template of our first part about tiophen derivatives (ANOVA test was used for the analysis of least significant differences of seedlings germination, see referee Lewkowski J., Malinowski Z., Matusiak A., Morawska M., Rogacz D., Rychter P., The Effect of New Thiophene-Derived Aminophosphonic Derivatives on Growth of Terrestrial Plants: A Seedling Emergence and Growth Test, Molecules, 2016, 21, 694).

In current manuscript (Part 2) we described results using only standard deviation of mean. With this respect, we clarified the text by removing the paragraph about ANOVA analysis.